# Convexity Certificates from Hessians

**Julien Klaus**
Friedrich Schiller University Jena
julien.klaus@uni-jena.de

**Niklas Merk**
Friedrich Schiller University Jena
niklas.merk@uni-jena.de

**Konstantin Wiedom**
Friedrich Schiller University Jena
konstantin.wiedom@uni-jena.de

**Sören Laue**
Technical University of Kaiserslautern
Data Assessment Solutions GmbH
laue@cs.uni-kl.de

**Joachim Giesen**
Friedrich Schiller University Jena
joachim.giesen@uni-jena.de

## Abstract

The Hessian of a differentiable convex function is positive semidefinite. Therefore, checking the Hessian of a given function is a natural approach to certify convexity. However, implementing this approach is not straightforward since it requires a representation of the Hessian that allows its analysis. Here, we implement this approach for a class of functions that is rich enough to support classical machine learning. For this class of functions it was recently shown how to compute computational graphs of their Hessians. We show how to check these graphs for positive semidefiniteness. We compare our implementation of the Hessian approach with the well-established disciplined convex programming (DCP) approach and prove that the Hessian approach is at least as powerful as the DCP approach for differentiable functions. Furthermore, we show for a state-of-the-art implementation of the DCP approach that, for differentiable functions, the Hessian approach is actually more powerful. That is, it can certify the convexity of a larger class of differentiable functions.

## 1 Introduction

Convex optimization forms the backbone of classical machine learning. Formulating machine learning problems as convex optimization problems is attractive because these problems can be solved globally and efficiently. Several optimization frameworks consisting of a modeling language and solver exist and are used in machine learning. Two examples of such frameworks are CVX [Grant and Boyd, 2014] and GENO [Laue *et al.*, 2019], which take different approaches.

CVX takes a formal specification of an optimization problem and transforms it into a normal form, such as a convex quadratic program (QP) or semidefinite program (SDP). The transformation is only possible if the specified problem is convex. Therefore, a convexity test is performed first in CVX. The test is based on a calculus for convex functions called disciplined convex programming (DCP) by Grant *et al.* [2006a], which takes advantage, for example, of the fact that the sum of convex functions and the positive scaling of a convex function are convex again. The convexity calculus requires a set of functions, called atoms, whose convexity has to be certified by other means.

GENO also takes the formal specification of an optimization problem but does not transform it into standard form. Instead, it uses the specification to generate a solver for the specified problem or

36th Conference on Neural Information Processing Systems (NeurIPS 2022).

problem class utilizing automatic (symbolic) differentiation. Thus, solvers can also be generated for non-convex problems. However, such solvers usually do not converge to a global optimum but only provide a local optimum. Therefore, to assess the quality of a solution, a convexity certificate calculated from the specification is also of interest to GENO.

For compiling a specification into a solver, GENO employs a matrix calculus [Laue *et al.*, 2018] for computing symbolic derivatives of matrix and tensor expressions. The matrix calculus can also be used to compute second order derivatives, that is, Hessians. A test for convexity can thus be reduced to a test of positive semidefiniteness of the Hessian, which certifies the convexity of a function. This is the approach that we take here. We design an algorithm for certifying positive semidefiniteness of matrix expressions. For certifying convexity, the algorithm is then applied to symbolic Hessians that are computed by a matrix calculus.

We start with a formally defined language of mathematical functions (see the supplementary material for its grammar). For expressions from this language, we generate computational graphs of their Hessians using a matrix calculus and check these graphs for positive semidefinitness. The computational graphs are small since they only contain symbols for parameter vectors and matrices, and not their numerical values. Our algorithm for certifying positive semidefiniteness runs in time linear in the size of the computational graph. It propagates positivity information of subexpressions to the root of the computational graph by using simple rules for positive semidefinite matrices. Non-trivial subexpressions can be certified as convex by matching them to known expression templates. Surprisingly, there is a single template that is powerful enough to certify the convexity of a large class of matrix expressions. Using this template, we show that our Hessian approach covers all differentiable functions with vector input that can be certified as convex by the DCP implementation within CVX. Furthermore, we provide classes of differentiable convex functions that can be certified by our approach, but cannot be certified as convex by CVX.

## 2 Related work

In general, certifying that a function is convex is known to be strongly NP-hard [Ahmadi *et al.*, 2013], even when the function is a multivariate polynomial of degree three and the domain is a bounded box [Ahmadi and Hall, 2020]. However, since certifying convexity is such an important issue in optimization and machine learning, there are many convexity proofs for specific problems, see Klibanov [1997] for an example. Unfortunately, these proofs are problem- and often even instance-specific and cannot be generalized. But, there are also generic approaches that we discuss in the following. Most generic approaches are either rule-based or analyze the Hessian.

**Rule-based approaches**  We have already mentioned the disciplined convex programming (DCP) approach taken by CVX [Grant and Boyd, 2008, 2014; Agrawal *et al.*, 2017], which has been ported from Matlab to other programming languages, like CVXPY [Diamond and Boyd, 2016] for Python, CVX for Julia [Udell *et al.*, 2014], or CVX for R [Fu *et al.*, 2020]. Any convex function that cannot be derived from the atomic convex functions by the DCP rule set cannot be certified as convex by the DCP approach. However, functions that have been certified as convex by other means can be added manually as atomic functions. Hence, over the course of the last few years, many convex functions have been added as atoms, for instance quadratic functions $x^\top A x$ for positive semidefinite matrices $A$, or the negative entropy function $x^\top \log(x)$. In our approach, we certify the convexity of these functions automatically.

Tawarmalani and Sahinidis [2005] describe a polyhedral branch-and-cut approach for finding a global optimum for non-convex optimization problems. They use rule-based convexity tests for decomposing a non-convex problem into a set of convex problems. Posypkin and Khamisov [2021] use interval arithmetic and a set of rules for determining the convexity of univariate functions, similar to the CVX approach. However, unlike CVX, their approach can only be applied to simple univariate functions. Similarly, Singh and Lucet [2021] analyze univariate piecewise polynomial functions. Fourer *et al.* [2010] summarize and generalize the convexity detection methods described by Schichl and Neumaier [2005] and Fourer and Orban [2010]. Their approach, traversing a function's computational graph and applying composition rules when convexity holds, is again similar to the DCP approach.

**Hessian based approaches**  Camino *et al.* [2003] use the software package NCAlgebra by Helton and de Oliveira [2000] for computing the Hessian of a function that is defined over matrices. Based

on a symbolic Cholesky decomposition, they check the non-negativity of the eigenvalues of the Hessian. If all eigenvalues are non-negative, the Hessian is positive semidefinite and thus implies convexity. However, this approach only works for very simple functions, i.e., only polynomials of matrices and their inverses. In these cases, the Hessian is always a quadratic function for which the symbolic Cholesky decomposition can be computed. This approach works for the function $x^\top A x$, but not for the negative entropy function $x^\top \log(x)$.

Using automatic differentiation for computing Hessians and then certifying convexity was proposed by Nenov *et al.* [2004]. However, the proposal does not include specific algorithms and has not been implemented yet.

**Other approaches** An approach that is neither rule-based nor based on analyzing the Hessian was proposed by Carmon *et al.* [2017], who established a relationship between the number of iterations needed for minimizing a function and its convexity. In general, if a function is convex, convergence rates can be estimated. Hence, if during the minimization process these convergence rates are violated, then one has found a certificate that this function is not convex. However, if convergence rates are satisfied during the optimization process no statement about convexity can be made. In general, if one starts close to a local minimum, then the function looks locally like a convex function to the minimization process, while globally, it does not need to be convex. Instead of looking at the convergence rates, one could modify this approach by computing the Hessian in each iteration and checking its smallest eigenvalue. If it is negative, then non-convexity can be certified. Again, convexity cannot be asserted by such an approach.

## 3 The DCP and the Hessian approach

This section gives a brief high-level overview of the DCP approach and our implementation of the Hessian approach. Details are provided in the following sections. The two approaches are algorithmically similar. Both use some a priori information that is propagated through expression DAGs (directed acyclic graphs) for the function or its Hessian, respectively. They differ in the form of the a priori information and in the rules that are used for propagating the information. Noteworthy, in contrast to the Hessian approach, the DCP approach also works for non-differentiable functions.

**DCP approach.** The DCP approach comprises a rule set and a set of atomic expressions, short atoms, that are already known to be convex. It applies to expressions that are recursively build from functions, constants, and variables. The expressions can be organized in an expression DAG whose inner nodes are function symbols and whose leaves are constants and variables as shown in Figure 1. The set of function symbols includes the atoms. Function symbols can be labeled as *convex*,

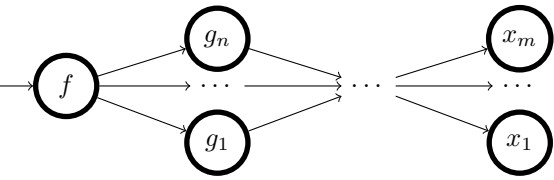

Figure 1: Abstract expression DAG for the function $f$. The leaves $x_i$ are either variables or constants, all other nodes $g_i$ and $f$ are abstract function symbols. The root of this sub-DAG is labeled by the function symbol $f$.

*concave*, *affine*, and *monotonously increasing (decreasing)*. Constants and variables can be labeled as *non-negative* or *non-positive*. The DCP rules are used to propagate label information from the leaves to the root of the expression DAG, which represents the whole expression. Therefore, if it is possible to propagate the label *convex* to the root, then the expression is proven to be convex.

**Hessian approach.** The Hessian of a twice differentiable function $f \colon \mathbb{R}^n \to \mathbb{R}$ is a quadratic form $H(f)$, that is, for $v \in \mathbb{R}^n$ the Hessian evaluates as $v^\top H(f) v$. The function $f$ is convex if $H(f)$ is positive semidefinite. Here, we assume that we can compute $H(f)$ in vectorized form, that is, in standard matrix language that does not make use of explicit indices. Our implementation works for a formally defined language for representing multivariate functions (see the supplemental material)

that allows to compute their Hessians in normalized vectorized form. By representing the Hessian by an expression DAG, the Hessian approach for computing convexity certificates propagates positivity and type information from the leaves of the DAG to the root by using a rule set that we introduce in Sections 4 and 5.

## 4 Rule sets

In this section we show that for a twice differentiable function the DCP rule set is implied by a set of positivity rules for the Hessian of the function. However, this is not enough to show that the DCP approach is implied by a positivity calculus, since this also requires that the atoms used in the DCP approach can be certified as convex by the positivity calculus. We discuss CVX's DCP atoms in Section 6.

The DCP rule set by Grant *et al.* [2006b] comprises the following rules for functions on $\mathbb{R}^n$:

1. $\sum_{i=1}^{m} \alpha_i f_i$ is convex if either $\alpha_i \geq 0$ and $f_i$ is convex, or $\alpha_i \leq 0$ and $f_i$ is concave, for all $i \in [m]$.

2. $f(g_1, g_2, \ldots, g_m)$ is convex if $f$ is convex and for each $g_i$ one of the following conditions holds:

    (a) $f$ is increasing in argument $i$ and $g_i$ is convex.
    (b) $f$ is decreasing in argument $i$ and $g_i$ is concave.
    (c) $g_i$ is affine or constant.

3. $f(g_1, g_2, \ldots, g_m)$ is concave if $f$ is concave and for each $g_i$ one of the following conditions holds:

    (a) $f$ is increasing in argument $i$ and $g_i$ is concave.
    (b) $f$ is decreasing in argument $i$ and $g_i$ is convex.
    (c) $g_i$ is affine or constant.

4. $f(g_1, g_2, \ldots, g_m)$ is affine if $f$ is affine and each function $g_i$ is affine.

Note that products of functions cannot be treated within the DCP framework, that is, expressions of the form $f_1 \cdot f_2$. Even when $f_1$ and $f_2$ are known to be affine, nothing can be said about the product.

Here, we only discuss certifying convexity, since concavity of a function $f$ can be certified by the convexity of $-f$. Hence, in the following, we do not consider the DCP Rule 3. We show that the DCP Rules 1 and 2 for twice differentiable functions are implied by positivity rules for their Hessians. DCP Rule 4 that asserts that $h$ is affine, which is a stronger property, can be addressed by adding the rules $0 \cdot M = M \cdot 0 = 0$ and $M + 0 = M$ to the positivity rules. More specifically, we have the following proposition.

**Proposition 1.** *For twice differentiable functions, the DCP rule set is implied by the following positivity rules for* $(n \times n)$*-matrices:*

1. *If $c \geq 0$ and $M \succeq 0$, then $c \cdot M \succeq 0$.*

2. *If $c \leq 0$ and $M \preceq 0$, then $c \cdot M \succeq 0$.*

3. *If $M_1 \succeq 0$ and $M_2 \succeq 0$, then $M_1 + M_2 \succeq 0$.*

4. *If $M \succeq 0$ and $A$ is an arbitrary $(m \times n)$ matrix, then $AMA^\top \succeq 0$.*

*Proof.* We exploit that a twice differentiable function $h$ is convex if its Hessian $H(h)$ is positive semidefinite (psd) and that it is concave if its Hessian $H(h)$ is negative semidefinite (nsd). For DCP Rule 1, we observe that $H(f_i) \succeq 0$ if $f_i$ is convex, and $H(f_i) \preceq 0$ if $f_i$ is concave. Hence, by Positivity Rules 1 and 2,

$$H(\alpha_i f_i) = \alpha_i H(f_i) \succeq 0$$

for all $i \in [m]$, and then by Positivity Rule 3,

$$H\Big( \sum_{i=1}^{m} \alpha_i f_i \Big) = \sum_{i=1}^{m} \alpha_i H(f_i) \succeq 0,$$

which certifies the convexity of $\sum_{i=1}^{m} \alpha_i f_i$.

For the remaining DCP Rules 2 and 4, let $h$ be recursively defined as $h = f(g_1, g_2, \ldots, g_m)$. The gradient of $h$ at $x$ can be written as

$$\nabla(h) = \sum_{i=1}^{m} \nabla(f)_i (g_1, g_2, \ldots, g_m) \cdot \nabla(g_i)$$

and its Hessian is given as

$$H(h) = \sum_{i,j=1}^{m} H(f)_{ij}(g_1, g_2, \ldots, g_m) \cdot \nabla(g_i)\nabla(g_j)^\top + \sum_{i=1}^{m} \nabla(f)_i(g_1, g_2, \ldots, g_m) \cdot H(g_i).$$

For DCP Rule 2a, we first argue separately that both sums for $H(h)$ are psd. The first sum can be written as $GH(f)G^\top$, where the columns of $G$ are the gradients $\nabla(g_i)$. Since the convexity of $f$ implies that $H(f) \succeq 0$ it follows from Positivity Rule 4 that the first sum for $H(h)$ is psd. For the second sum, we use that $f$ is increasing in its $i$-th argument, which implies that $\nabla(f)_i \geq 0$, and that the $g_i$ are convex, which implies that $H(g_i) \succeq 0$. Hence, by Positivity Rule 1 all terms in the second sum are psd, which together with Positivity Rule 3 implies that also the second sum for $H(h)$ is psd. Thus, we can conclude from Positivity Rule 3 that $H(h) \succeq 0$, which certifies the convexity of $h$. The claims for DCP Rules 2b, 2c and 4 follow similarly. $\qquad\square$

## 5  Implementing the Hessian approach

For implementing the Hessian approach, we need to compute a representation of the Hessian that is amenable to analysis. The matrix calculus by Laue *et al.* [2018, 2020] computes derivatives of vectorized expressions again in vectorized form without explicit indices. Here, we employ computational graphs for the expressions of second derivatives. That is, Hessians, computed by the matrix calculus. We normalize these computational graphs into expression DAGs (directed acyclic graphs) that contain each subexpression exactly once, see Figures 2, 3 and 4 for examples of normalized expression DAGs.

In the following, we demonstrate our implementation of the Hessian approach using illustrative examples before we summarize the algorithm that underlies our implementation of the Hessian approach.

### 5.1  A generic example

As a first generic example we discuss the ordinary least squares regression problem

$$\min_w \|Xw - y\|_2^2 = \min_w (Xw - y)^\top (Xw - y),$$

where $X \in \mathbb{R}^{m \times n}$ is a data matrix, $y \in \mathbb{R}^m$ is a label vector, and $w \in \mathbb{R}^n$ is the parameter vector that needs to be optimized. Using matrix calculus, the Hessian for the objective function of this problem can be computed in vectorized form as $2 \cdot X^\top X$. An expression DAG for the Hessian is shown in Figure 2. Our implementation of the Hessian approach computes positivity information for each node

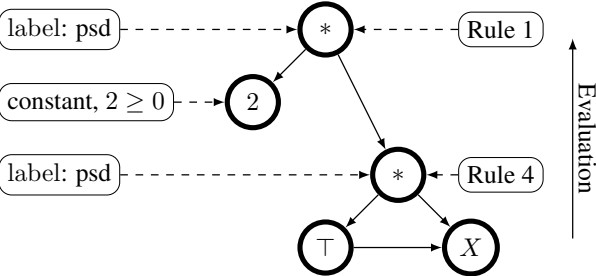

Figure 2: Illustration of the Hessian approach on the normalized expression DAG for the Hessian of the ordinary least squares loss function.

of the DAG in a bottom-up strategy from the leaves to the root. There are positivity rules that apply to the two multiplication nodes of the DAG, namely first Rule 4 for the expression $X^\top X$ and then Rule 1 for the expression $2 \cdot X^\top X$, where it is already known that $X^\top X$ is psd.

Let us compare the Hessian approach to the DCP approach for certifying the convexity of the ordinary least squares problem. The DAG for the ordinary least squares objective function is shown in Figure 3. The DCP approach traverses this DAG in a bottom-up fashion. Starting from the leaves, the matrix-vector product $Xw$ is affine by definition and thus the corresponding multiplication node is labeled affine, similarly the subtraction node for $Xw - y$ is labeled affine, since the addition/subtraction of affine functions is affine again, and the transposition node in $(Xw - y)^\top$ is labeled affine, since the transposition preserves this property. However, in general nothing can be said about the product of affine functions, which means that the standard DCP rules are not enough to label the root of the DAG. Typically, the problem is dealt with by adding a new atomic function to the DCP atoms that squares its input.

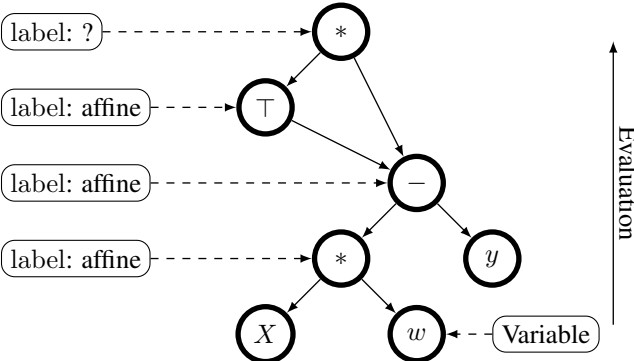

Figure 3: Illustration of the DCP approach on the normalized expression DAG for the ordinary least squares loss function.

## 5.2 Propagating information about subexpressions

A second instructive example is the univariate function $f(x) = x \log(x)$ that neither the Hessian nor the DCP approach can certify as convex without information beyond the rules. However, the information required by the Hessian approach is easier to provide on the language level. The Hessian of $f(x)$ is the expression $1/x$, which, in general, can be positive, negative, or even undefined at $0$. However, we already know that $x \in (0, \infty)$, because the domain of the logarithm is the positive reals. This information is enough to decide the positivity of the Hessian. Hence, we can exploit positivity information about the elementary functions that are supported by our formal language. Here are some additional rules

$$\log(x) \ \Rightarrow \ x \in (0, \infty), \ \sqrt{x} \ \Rightarrow \ x \in [0, \infty),$$

and $\exp(x), \arccos(x), \log(1 + x), \|x\| \ \geq \ 0$.

Let us compare this again to the DCP approach that directly works on the expression $x \log(x)$. Here, the problem is at the root node that multiplies the affine function $x$ with the concave function $\log(x)$. Since, in general, nothing can be said about the product of an affine function and a concave function, it is not possible to certify the convexity of $x \log(x)$ from the DCP rules. Indeed, CVX adds the entropy function $-x^\top \log(x)$ as an atom that has been certified as concave by other means like analyzing its second order derivative.

Another instructive example is the univariate function $\log(1 + \exp(x))$ whose Hessian is

$$\frac{\exp(x)}{1 + \exp(x)} \left( 1 - \frac{\exp(x)}{1 + \exp(x)} \right).$$

The Hessian can be certified positive definite by propagating the known positivity information for the exponential function through the normalized expression DAG for the Hessian, as can be seen in Figure 4. CVX adds this function as an atom named `logistic`, probably because the derivative of

this function is the logistic function $1/(1 + \exp(-x))$. Note that the naming problem does not exist in the Hessian approach.

A fourth example is the power function $x^p$. The Hessian of the power function is $p(p-1)x^{p-2}$. Since $p(p-1) \geq 0$ for $p \geq 1$ and $p \leq 0$ we can certify the convexity of the power function from its Hessian if $p = 2n, n \in \mathbb{N}$, or $((p \geq 1) \vee (p < 0)) \wedge (x > 0)$, or $((p = 1) \vee (p = 2k, k \in \mathbb{Z})) \wedge (x < 0)$. Furthermore, we can certify the convexity of $-x^p$ from its Hessian if $(0 < p < 1) \wedge (x \geq 0)$. That is, the Hessian approach can certify the convexity of power functions from constraints on $x$ and information about the power parameter $p$, whereas the DCP approach needs atoms for the different cases.

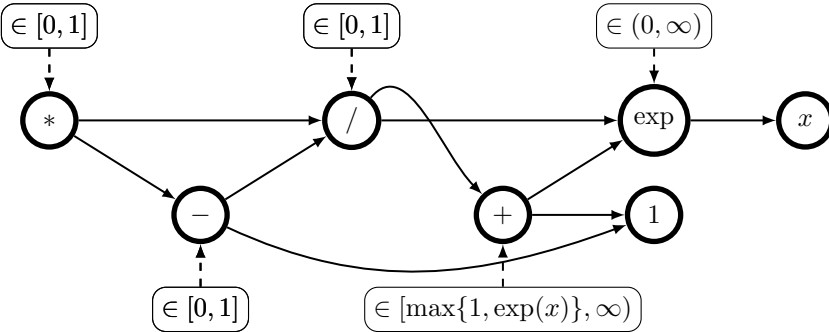

Figure 4: Propagating positivity information from the leaves to the root of the normalized expression DAG for the Hessian of the logistic function.

In general, the Hessian approach can exploit derived and user provided information about variables, parameters, and, more generally, subexpressions. However, the next example shows that this is not enough.

## 5.3 A psd expression template

Our fifth example is the CVX atom `log_sum_exp` that computes $\log\big(\operatorname{sum}(\exp(x))\big)$. Its Hessian is given, again in vectorized form, as

$$\Big(\operatorname{diag}\big(\exp(x)\big) - \exp(x)\exp(x)^\top / \operatorname{sum}\big(\exp(x)\big)\Big) / \operatorname{sum}\big(\exp(x)\big).$$

Both the first and the second term are readily certified as positive, but their difference is not, because in general, nothing can be said about the difference of two psd matrices. However, we will show that the Hessian matches the psd expression template from Proposition 2, see also the Figure to the right. The template can be used to certify many expressions as convex. Whenever a subexpression of a larger expression DAG matches the template, we can label the subexpression DAG as psd and propagate this information within the larger DAG. Indeed, adding this template to our rule set is powerful enough to cover all differentiable CVX atoms with vector input, and thus by Proposition 1 all differentiable functions with vector input, that can be certified as convex by the DCP implementation within CVX.

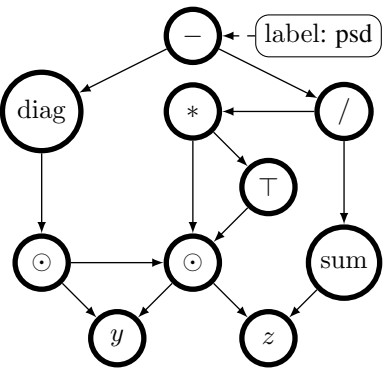

**Proposition 2.** *For $y \in \mathbb{R}^n$ and $z \in \mathbb{R}^n_{\geq 0}$, all matrices of the following form are psd:*

$$diag(y \odot z \odot y) - (y \odot z)(y \odot z)^\top / sum(z).$$

*Proof.* We have to show that $v^\top A v \geq 0$ for all $v \in \mathbb{R}^n$ for the template matrix $A$. It turns out that $v^\top A v$ is the variance of some random variable. We compute that

$$
\begin{aligned}
v^\top(\operatorname{diag}(y \odot z \odot y) &- (y \odot z)(y \odot z)^\top/\operatorname{sum}(z))v \\
&= \operatorname{sum}(z \odot (y \odot v)^2) - \operatorname{sum}(z \odot (y \odot v))^2/\operatorname{sum}(z) \\
&= \operatorname{sum}(z) \cdot \Big(\operatorname{sum}((z/\operatorname{sum}(z)) \odot (y \odot v)^2) - \operatorname{sum}((z/\operatorname{sum}(z)) \odot y \odot v)^2\Big) \\
&= \operatorname{sum}(z) \cdot \operatorname{var}(V_y) \geq 0,
\end{aligned}
$$

where $\operatorname{var}(V_y)$ is the variance of the random variable $V_y$ that takes the value $y_i v_i$ with probability $z_i/\operatorname{sum}(z)$ for $i \in [n]$. $\qquad\square$

Note that the Hessian of the CVX atom `log_sum_exp` matches the template if $z$ is instantiated by $\exp(x)/\operatorname{sum}(\exp(x))$ and $y$ by vector(1).

### 5.4 Summary of the Hessian approach

Algorithm 1 that implements the Hessian approach combines the individual steps mentioned above. Its input is the Hessian of a given expression computed as a normalized expression DAG. In the algorithm, we encode positivity and negativity information by intervals as follows: for scalar expressions, the interval encodes (a superset of) the domain of the expression, for vector expressions, the interval encodes (a superset of) the domains of all vector entries, and for matrix expression, any interval in $[0, \infty)$ encodes psd information and any interval in $(-\infty, 0]$ encodes nsd information.

---

**Algorithm 1** Certify convexity of a Hessian

---
1: $v \leftarrow \textsc{RootOf}(\text{DAG})$
2: $I_v \leftarrow \textsc{DetermineInterval}(v)$
3: **if** $I_v \subseteq [0, \infty)$ **then**
4:     **return** true
5: **else**
6:     **return** false
7: **end if**

---

The main work in Algorithm 1 is delegated to the subroutine $\textsc{DetermineInterval}$ that is implemented in Algorithm 2. The subroutine recursively processes the normalized expression sub-DAG rooted at some vertex $v$ from the leaves to the node $v$. The intervals $I_v$ at leaf nodes always either encode known or user-provided positivity information about variables or parameters or are set to $(-\infty, \infty)$. At any node, the information about the intervals associated with its children is combined, using the psd rule set and standard interval arithmetic, into an interval for the node by the subroutine $\textsc{CombineIntervals}$. Exceptions from this rule are multiplication nodes at which Rule 4 might apply and subtraction nodes, where the psd template from Proposition 2 might apply. Therefore, we check at each node if a simple template for Rule 4 or our psd template applies. Matching the templates are simple instances of tree matching problems that are implemented in the subroutine $\textsc{MatchTemplate}$. Here, we encode a match with the templates by the interval $[0, \infty)$.

Since the psd expression template and the rules from Proposition 1 only require checking substructures of constant depth (up to depth five for the psd template), the running time of the algorithm is linear in the size of the expression DAG of the Hessian.

## 6 Atoms of CVX's implementation of the DCP approach

In addition to the rule set, the DCP approach requires functions (atoms) that are already known to be convex. Here, we show that all twice differentiable atoms with vector input that can be certified as convex by the DCP implementation within CVX can also be certified by our implementation of the Hessian approach. In the next section we provide examples of convex functions that are not certified as convex by CVX, but can be certified by our Hessian approach.

Among CVX's DCP atoms are standard convex univariate functions like `exp`, `neg_log`, `neg_sqrt`, and `square` that we discuss in the supplemental material. Multivariate DCP atoms in CVX are

---

**Algorithm 2** Compute the positivity interval for a node $v$

---

1: **procedure** DETERMINEINTERVAL($v$)
2: **if** $v$.leaf = true **then**
3:     **return** $I_v$
4: **end if**
5: **if** MATCHTEMPLATE($v$) **then**
6:     **return** $[0, \infty)$
7: **else**
8:     $I_l \leftarrow$ DETERMINEINTERVAL(LEFTCHILD($v$))
9:     $I_l \leftarrow$ DETERMINEINTERVAL(RIGHTCHILD($v$))
10:    **return** COMBINEINTERVALS($I_l, I_r$)
11: **end if**
12: **end procedure**

---

$\texttt{sum}(x)$, that is, the function $\sum_{i=1}^{n} x_i$, and quadratic forms $\texttt{quad\_form}(x, A)$, that is, $x^\top A x$ for a psd matrix $A$. Both functions are readily certified as convex by their Hessians. A more interesting atom is $\texttt{inv\_prod}(x)$ that computes $\left( \prod_{i=1}^{n} x_i \right)^{-1}$. In vectorized notation this function can be written as $f(x) = 1/\exp\big(\text{sum}(\log(x))\big)$ and its vectorized Hessian is given as

$$f(x) \cdot \big( \text{vector}(1) \oslash x)(\text{vector}(1) \oslash x)^\top + \text{diag}(\text{vector}(f(x)) \oslash (x \odot x)\big),$$

where $\text{vector}(1)$ is the all ones vector, $\odot$ and $\oslash$ denote elementwise multiplication and divison, respectively. It follows from the positivity of $f(x)$ and Positivity Rules 1 and 4 that the first term in the sum for the Hessian is psd, and from the positivity of $f(x)$ and the positivity of the entries of $x \odot x$ it follows that also the second term is psd. Hence, the Hessian is psd by Positivity Rule 3, which certifies the convexity of $f(x)$.

CVX also contains four atoms (combinations of the operators sum, log, and exp) that superficially look similar to $\texttt{inv\_prod}$ but cannot be certified directly from the positivity calculus. However, for these problems, the Hessian is the difference of two psd matrices that can be matched to the template expression from Proposition 2. We have already discussed the CVX atom $\texttt{log\_sum\_exp}$. The three remaining atoms are the harmonic mean, $p$-norms, and the geometric mean.

**Negative harmonic mean** The negative harmonic mean $\texttt{neg\_harmonic\_mean}(x)$ for $x \in \mathbb{R}_+^n$ is defined as $-n/\sum_{i=1}^{n} x_i^{-1}$. It can be written in vectorized notation as

$$-1/\text{sum}\big(\text{vector}(1) \oslash x)\big) =: f(x).$$

Its Hessian, which computes to

$$2 \cdot f(x)^2 \Big( \text{diag}\big(\text{vector}(1) \oslash (x \odot x \odot x)\big) + f(x) \cdot \ (\text{vector}(1) \oslash (x \odot x))(\text{vector}(1) \oslash (x \odot x))^\top \Big),$$

is matched by the psd expression template if both $y$ and $z$ are instantiated by $\text{vector}(1) \oslash x$. Note that $1/\text{sum}(z)$ in the template becomes $-f(x) = 1/\text{sum}\big(\text{vector}(1) \oslash x)\big)$ in the instantiation.

**$p$-norms** The $p$-norm $\|x\|_p = \left( \sum_{i=1}^{n} |x_i|^p \right)^{1/p}$ of $x \in \mathbb{R}^n$ for $p > 1$ reads in vectorized notation as

$$\text{sum}\big(\exp(p \cdot \log(s(x) \odot x))\big)^{1/p} = \text{sum}(f(x))^{1/p},$$

where $s(x)$ is the sign vector of $x$, that is, we have $s(x) \odot s(x) = s(x) \oslash s(x) = \text{vector}(1)$. The Hessian of the $p$-norm can be computed as

$$(p-1) \cdot \text{sum}((f(x))^{1/p-1} \cdot \text{diag}(f(x) \oslash (x \odot x)) - (p-1) \cdot \text{sum}(f(x))^{1/p-2} \cdot (f(x) \oslash x)(f(x) \oslash x)^\top.$$

The Hessian matches the psd expression template if $y$ is instantiated by $\text{vector}(1) \oslash x$ and $z$ by $f(x)$. It also follows that the $p$-norm is concave for $0 < p < 1$, since the negated $p$-norm is convex for these values of $p$.

**Negative geometric mean** The negative geometric mean $\texttt{neg\_geo\_mean}(x, p)$ is given as

$$-\left(\prod_{i=1}^{n} x_i^{p_i}\right)^{1/\mathrm{sum}(p)},$$

where $p \in \mathbb{R}_+^n$ is a parameter vector. It reads in vectorized form as

$$f(x) = -\exp(\mathrm{sum}(p \odot \log(x)))^{1/\mathrm{sum}(p)} \leq 0.$$

Its Hessian

$$f(x) \cdot \left((p \oslash x)(p \oslash x)^\top / \mathrm{sum}(p)^2 - \mathrm{diag}\big((p \oslash x \oslash x)/\mathrm{sum}(p)\big)\right),$$

matches the psd expression template if $y$ is instantiated by $p$ and $z$ is instantiated by $\mathrm{vector}(1) \oslash x$.

## 7 Beyond CVX's atoms

There are two main classes of functions that cannot be treated by DCP. The first are products of two non-constant expressions such as $x \exp(x)$ for $x \geq 0$. The second are compositions that are not following DCP's Rules 2, 3 or 4 such as $\texttt{neg\_entr}(\cosh(x)) = \cosh(x) \log(\cosh(x))$, where $\texttt{neg\_entr}(x) = x \log(x)$ is convex but not increasing, and thus composing it with the convex function $\cosh(x)$, DCP Rule 2a does not apply here. Both examples so far do not need the template expression from Proposition 2 to be certified convex by the Hessian approach. This is different for the multivariate function

$$\Big(\sum_{i=1}^{n} \exp(x_i)\Big) \log\Big(1 + \sum_{i=1}^{n} \exp(x_i)\Big),$$

which can be expressed in vectorized form as

$$\mathrm{sum}\big(\exp(x)\big)\log\big(1 + \mathrm{sum}\big(\exp(x)\big)\big).$$

Its Hessian is the sum of three matrices

$$\left(\log\big(1 + \mathrm{sum}(\exp(x))\big) + \frac{\mathrm{sum}(\exp(x))}{(1 + \mathrm{sum}(\exp(x)))^2}\right) \cdot \mathrm{diag}(\exp(x)) + \frac{2 \cdot \exp(x)\exp(x)^\top}{1 + \mathrm{sum}(\exp(x))}$$

$$+ \left(\frac{\mathrm{sum}(\exp(x))}{1 + \mathrm{sum}(\exp(x))}\right)^2 \cdot \Big(\mathrm{diag}(\exp(x)) - \exp(x)\exp(x)^\top / \mathrm{sum}(\exp(x))\Big),$$

where the last matrix matches the template expression from Proposition 2, up to a positive prefactor if $y$ is instantiated by $\mathrm{vector}(1)$ and $z$ by $\exp(x)$. The first two matrices can be certified as psd directly from the positivity rules.

## 8 Conclusions

We have presented the first generic implementation of the Hessian approach for certifying convexity and shown that it complements the well-established disciplined convex programming approach. Neither approach is better than the other. Both have complementary strengths and weaknesses. The DCP approach also works for non-differentiable functions but needs a new symbol for every new atom, whereas our implementation of the Hessian approach works on a formal language that is close to natural problem formulations in textbooks and rich enough to express many classical machine learning problems, including problems not found in standard libraries like scikit-learn [Pedregosa *et al.*, 2011]. Furthermore, new DCP atoms also need to be certified at some point, and the Hessian approach can be used to certify some of these new atoms.

**Acknowledgments** This work was supported by the Carl Zeiss Foundation within the project "Interactive Inference" and by the Ministry for Economics, Sciences and Digital Society of Thuringia (TMWWDG) under the framework of the Landesprogramm ProDigital (DigLeben-5575/10-9).

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
