# Convexity Certificates from Hessians (Supplementary Material)

Here, we (1) provide the grammar for the formal language of mathematical expressions to which our certification algorithm is applied, (2) we provide more algorithmic details about our implementation of the Hessian approach, (3) we show that our implementation of the Hessian approach can also certify the remaining differentiable CVX atoms with vector input, which we could not discuss in the main paper because of space constraints, and (4) we provide more examples of differentiable functions that can be certified by the Hessian approach but are missing from CVX's DCP implementation.

## 1 Grammar for mathematical expressions

The formal language for mathematical expressions to which our certification algorithm is applied is specified by the grammar depicted in Figure 1. The language is rich enough to cover all the examples in the main paper and this supplement.

$\langle expr \rangle$      ::= $\langle term \rangle$ | $\langle term \rangle$ + $\langle term \rangle$ | $\langle term \rangle$ - $\langle term \rangle$

$\langle term \rangle$      ::= [-] $\langle factor \rangle$ | $\langle factor \rangle$ [.] * $\langle factor \rangle$ | $\langle factor \rangle$ [.] / $\langle factor \rangle$

$\langle factor \rangle$      ::= $\langle atom \rangle$ ['] [[.] ^ $\langle factor \rangle$]

$\langle atom \rangle$      ::= number | function ( $\langle expr \rangle$ ) | variable

Figure 1: EBNF Grammar for mathematical expressions supported by our approach. In this grammar, *number* is a placeholder for an arbitrary floating point number, *variable* is a placeholder for variable names starting with a Latin character and *function* is a placeholder for the supported elementary differentiable functions like $\exp, \log$ and *sum*. Here, $'$ is used for transposition and a preceding . introduces an elementwise operation.

Here are some examples from the language (the fist example uses a transposition and the fifth and seventh example use elementwise operations):

**2-norm** $\|Xw - y\|^2$:    (X*w-y)'*(X*w-y)

**logistic** $\log(1 + \exp(x))$:    log(1+exp(x))

**quadratic** $x^2$:   `x^2`

**relative entropy** $x\log(x/y)$:   `x*log(x/y), x>0, y>0`

**logistic regression** $\sum_{i=1}^{n}\log(\exp(y_i \cdot x_i^\top w)+1)$:   `sum(log(exp(-y.*(X*w))+vector(1)))`

**inverse product** $(\prod_{i=1}^{n} x_i)^{-1}$:   `1/exp(sum(log(x)))`

**harmonic mean** $1/((\sum_{i=1}^{n} 1/x_i)/n)$:   `n/sum(x.^(-1)),n>0,x>0`

## 2 Algorithmic details of the Hessian approach

Our implementation of the Hessian approach works on vectorized and normalized expression DAGs (directed acyclic graphs) for Hessians that contain every subexpression exactly once. Our formal input language is vectorized, which means expressions are expressed without indices but not normalized. Using a matrix calculus implementation, we can compute vectorized Hessians that are again not normalized. Therefore, we need to normalize such expressions.

We illustrate the normalization on the expression `x*A'*x + exp(x*A'*X)`, which represents the function $xA^\top x + \exp(xA^\top x)$. First, we parse the expression into a standard expression tree. The expression tree for our example is shown in Figure 2.

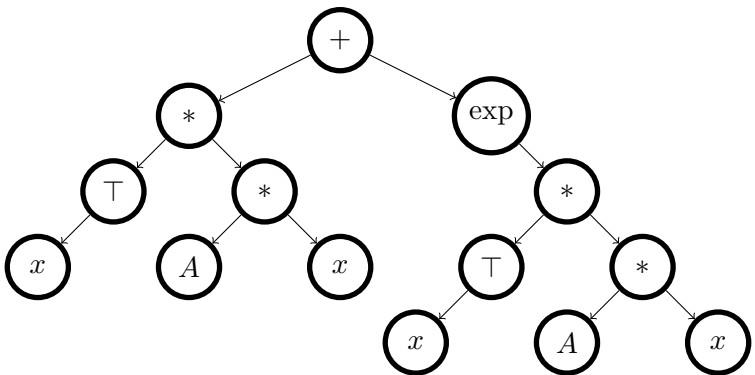

Figure 2: Expression tree for expression `x*A'*x + exp(x*A'*X)`.

Using a common subexpression elimination [Aho et al.: Compilers: Principles, Techniques, and Tools, 2013] we identify the common subexpressions $x$, $x^\top$, $A$, $Ax$, $x^\top Ax$ that appear more than once. Rearranging these subexpressions in the expression tree results in the normalized DAG that is shown in Figure 3. Every node in the normalized DAG represents a unique subexpression.

The normalized DAG then serves as the data structure on which the Hessian approach for certifying convexity operates. The Hessian approach traverses the DAG in post-order and attempts to label the nodes (subexpressions) of the DAG . This way, positivity information is propagated from the leaves of the DAG to the root.

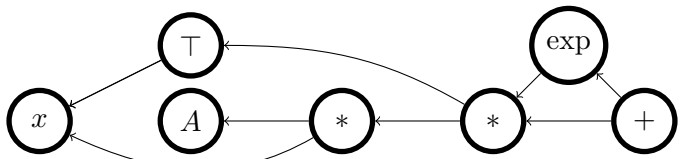

Figure 3: Normalized expression DAG after common subexpression elimination of the subexpressions `x*A'*x + exp(x*A'*X)`. The leaves of the DAG are the nodes labeled by the variable $x$ and the parameter matrix $A$, respectively. The root of the tree is the right most node (labeled by +) and represents the whole expression.

# 3   CVX atoms

Due to space constraints, we could not discuss all differentiable CVX atoms in the main paper. Here, we show that also the remaining atoms can be certified as convex by our implementation of the Hessian approach. We start with standard univariate functions before we discuss the remaining multivariate functions.

**Univariate functions.**   CVX's univariate DCP atoms are standard convex univariate functions like `exp`, `neg_log`, `neg_sqrt`, and `square`. For certifying these atoms as convex, the Hessian approach makes use of the information that $\exp(x) > 0$, that $\log(x)$ implies $x > 0$, and that $\sqrt{x}$ implies $x \geq 0$. This works analogously for the atom $\texttt{log1p}(x) = \log(1+x)$. We have already discussed power functions and the negative entropy function `neg_entr` in the examples (see main paper). The atom `inv_pos` that encodes the function $1/x$ for $x > 0$ can be certified as convex from its Hessian by using the constraint $x > 0$.

**Remaining multivariate functions.**   The only remaining twice differentiable multivariate functions are $\texttt{sum}(x) = \sum_{i=1}^{n} x_i$ and $\texttt{sum\_squares}(x) = \sum_{i=1}^{n} x_i^2$, which are straightforwardly verified by the Hessian approach.

**Exceptions.**   Similar, and related to the negative entropy function, are the relative entropy (`rel_entropy`) $x \log(x/y)$ and the Kullback-Leibler divergence (`kl_div`) $x \log(x/y) - x + y$ for $y > 0$. Together with the atom $\texttt{quad\_over\_lin}(x, y) = \sum_{i=1}^{n} x_i^2/y$ for $y > 0$, these are the only twice differentiable atoms not tractable with our approach, as they cannot be expressed with a single vector input.

# 4   More examples of convex functions not covered by CVX

In the last section we saw that the expression template derived in the main paper suffices to certify all twice differentiable atoms with vector input of CVX. Here, we collect a few

more examples of convex functions that can be certified by the Hessian approach using the expression template but are not feasible for DCP.

## 4.1   Examples matched by the basic expression template

The following examples of multivariate, vectorized functions can be matched to the expression template from Proposition 2 (main text).

(1)

$$f(x) = \sqrt{\sum_{i=1}^{n} \cosh(x_i) \cdot \log\left(\sum_{i=1}^{n} \cosh(x_i)\right)} = \text{sqrt}(\text{sum}(\cosh(x)))\log(\text{sum}(\cosh(x)))$$

with its Hessian

$$\frac{1}{2\sqrt{\text{sum}(\cosh(x))}} \left(\frac{\log(\text{sum}(\cosh(x)))}{2} + 1\right)$$

$$\left(\text{diag}(1 \oslash \cosh(x) + \cosh(x)) + \text{diag}(y \odot z \odot y) - (y \odot z)(y \odot z)^\top / \text{sum}(z)\right)$$

$$+ \frac{\sinh(x)\sinh(x)^\top}{2\,\text{sum}(\cosh(x))^{3/2}},$$

where $y = \sinh(x) \oslash \cosh(x)$ and $z = \cosh(x)$.

The next examples require a restriction of the domain by the user.

(2)

$$f(x) = \|x\|_2 \cdot \log\|x\|_2 = \text{norm2}(x)\log\big(\text{norm2}(x)\big) = \text{sum}(x \odot x)^{1/2}\log\left(\text{sum}(x \odot x)^{1/2}\right)$$

with $\|x\|_2 \geq 1$ and Hessian

$$\frac{1}{\text{norm2}(x)}\text{diag}(1) + \frac{\log\big(\text{norm2}(x)\big)}{\text{norm2}(x)} \left(\text{diag}(y \odot z \odot y) - (y \odot z)(y \odot z)^\top / \text{sum}(z)\right),$$

where $y = \text{vector}(1) \oslash x$ and $z = x \odot x$.

(3)

$$f(x) = \sqrt{\sum_{i=1}^{n} \exp(x_i) \cdot \log\left(\sum_{i=1}^{n} \exp(x_i)\right)} = \text{sqrt}(\text{sum}(\exp(x)))\log(\text{sum}(\exp(x)))$$

with $\sum_{i=1}^{n} \exp(x_i) \geq 1$ and Hessian

$$\frac{1}{\sqrt{\text{sum}(\exp(x))}} \left(\frac{\log(\text{sum}(\exp(x)))}{4} + 1\right) \text{diag}(\exp(x))$$

$$+ \frac{\log(\text{sum}(\exp(x)))}{4\sqrt{\text{sum}(\exp(x))}} \left(\text{diag}(y \odot z \odot y) - (y \odot z)(y \odot z)^\top / \text{sum}(z)\right),$$

where $y = \text{vector}(1)$ and $z = \exp(x)$.

(4)

$$f(x) = \left(1 + \sum_{i=1}^{n} \exp(x_i)\right) \log \left(\sum_{i=1}^{n} \exp(x_i)\right) = (1 + \mathrm{sum}(\exp(x)))\log(\mathrm{sum}(\exp(x)))$$

with $\sum_{i=1}^{n} \exp(x_i) \geq 1$ and Hessian

$$\log(\mathrm{sum}(\exp(x)))\,\mathrm{diag}(\exp(x)) + \frac{2\exp(x)\exp(x)^{\top}}{\mathrm{sum}(\exp(x))}$$

$$+ \frac{1 + \mathrm{sum}(\exp(x))}{\mathrm{sum}(\exp(x))}\left(\mathrm{diag}(y \odot z \odot y) - (y \odot z)(y \odot z)^{\top}/\mathrm{sum}(z)\right),$$

where $y = \mathrm{vector}(1)$ and $z = \exp(x)$.

## 4.2 Examples without expression template

There are also examples, including univariate functions, which do not require the use of an expression template but can be verified directly.

(1)

$$f(x) = \left(\sum_{i=1}^{n} \cosh(x_i)\right) \log \left(\sum_{i=1}^{n} \cosh(x_i)\right) = \mathrm{sum}(\cosh(x))\log(\mathrm{sum}(\cosh(x)))$$

with its Hessian

$$\left(\log(\mathrm{sum}(\cosh(x))) + 1\right)\mathrm{diag}(\cosh(x)) + \frac{\sinh(x)\sinh(x)^{\top}}{\mathrm{sum}(\cosh(x))}.$$

The following examples need a restriction of the domain by the user.

(2)

$$f(x) = \sum_{i=1}^{n} \exp(x_i) \log(x_i) = \mathrm{sum}(\exp(x) \odot \log(x))$$

with $x \geq \mathrm{vector}(1)$ and Hessian

$$\mathrm{diag}\left(\exp(x) \odot \log(x) \odot \left(\mathrm{vector}(1) + \mathrm{vector}(1) \oslash x - (\mathrm{vector}(1) \oslash x) \odot (\mathrm{vector}(1) \oslash x)\right)\right).$$

Note that it is psd, as $\mathrm{vector}(1) \oslash x \in [0, 1]^n$ and the function $z \mapsto z - z^2$ is non-negative on $[0, 1]$.

(3)

$$f(x) = \sum_{i=1}^{n} x_i \log(1 + \exp(x_i)) = \mathrm{sum}(x \odot \log(\mathrm{vector}(1) + \exp(x)))$$

with $x \in \mathbb{R}^n_{\geq 0}$ and Hessian

$$\mathrm{diag}\left(\frac{2\exp(x)}{1+\exp(x)}\right)$$
$$+\mathrm{diag}\left(x \odot \left(\frac{\exp(x)}{1+\exp(x)} - \frac{\exp(x)}{1+\exp(x)} \odot \frac{\exp(x)}{1+\exp(x)}\right)\right).$$

Here, the second term is psd by the same argument as above.

(4)
$$f(x) = \sum_{i=1}^{n} \exp(x_i)\log(\cosh(x_i)) = \mathrm{sum}(\exp(x) \odot \log(\cosh(x)))$$

with $x \in \mathbb{R}^n_{\geq 0}$ and Hessian

$$\mathrm{diag}\Big(\exp(x) \odot \Big(\log(\cosh(x)) + 2\sinh(x) \oslash \cosh(x) + \mathrm{vector}(1) \oslash (\cosh(x) \odot \cosh(x))\Big)\Big).$$

(5)
$$f(x) = \sum_{i=1}^{n} \exp(x_i) - \frac{1}{4}\exp(2x_i) = \mathrm{sum}(\exp(x) - 1/4\exp(2x))$$

with Hessian $\mathrm{diag}(\exp(x) - \exp(2x))$, which is psd for $x \in \mathbb{R}^n_{\leq 0}$ and nsd for $x \in \mathbb{R}^n_{\geq 0}$.

(6)
$$f(x) = \left(\sum_{i=1}^{n} \exp(x_i)\right)\log\left(\sum_{i=1}^{n} \exp(x_i)\right) = \mathrm{sum}(\exp(x))\log(\mathrm{sum}(\exp(x)))$$

with $\sum_{i=1}^{n} \exp(x_i) \geq \exp(-1)$. Its Hessian reads

$$\big(\log(\mathrm{sum}(\exp(x))) + 1\big)\mathrm{diag}(\exp(x)) + \frac{\exp(x)\exp(x)^\top}{\mathrm{sum}(\exp(x))}.$$

Here are some univariate functions which can be classified by the Hessian approach once the domain is restricted:

(1) $f(x) = x^a\exp(x)$ for $0 < a < 1$ and $x \geq 1$ or $a < 0$ and $x > 0$. Note that for $a \geq 1$, $x^a\exp(x)$ can be expressed as $\frac{1}{a^a}\left(\mathrm{xexp}(x/a)\right)^a$ with the atom $\mathrm{xexp}(x) = x \cdot \exp(x)$.

(2) $f(x) = x^b\log(x)$ for $b > 1$ and $x \geq 1$.

(3) $f(x) = x\cosh(x)$ for $x \geq 0$.

## 4.3 Generalized psd expression template

To support even more convex functions, we can generalize the expression template as shown in Figure 4.

**Proposition 3.** *For $y \in \mathbb{R}^n$, $z \in \mathbb{R}^n_{\geq 0}$, $a \geq 1$ and $b \geq 0$, all matrices of the following form are psd:*

$$diag(y \odot z \odot y) - (y \odot z)(y \odot z)^\top / (a(b + sum(z))).$$

*Proof.* We can use the template expression from Proposition 2 (main text) to show that the above expression is also psd. To do so, we split the first summand of

$$
\text{diag}(y \odot z \odot y) - \frac{(y \odot z)(y \odot z)^\top}{a(b + \text{sum}(z))}
$$

$$
= \frac{a-1}{a} \text{diag}(y \odot z \odot y) + \frac{1}{a}\left( \text{diag}(y \odot z \odot y) - \frac{(y \odot z)(y \odot z)^\top}{b + \text{sum}(z)} \right)
$$

$$
= \frac{a-1}{a} \text{diag}(y \odot z \odot y) + \frac{1}{a}\left( \frac{b}{b + \text{sum}(z)} \text{diag}(y \odot z \odot y) \right.
$$

$$
\left. + \frac{\text{sum}(z)}{b + \text{sum}(z)} \left( \text{diag}(y \odot z \odot y) - (y \odot z)(y \odot z)^\top / \text{sum}(z) \right) \right),
$$

and note that the resulting first two summands are diagonal matrices with non-negative entries, hence psd, and the last summand was shown to be psd in Proposition 2 (main text). □

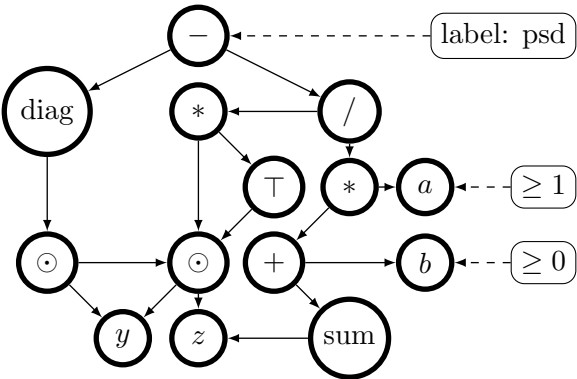

Figure 4: General template for determining the psd property of subtractions.

In addition, the template allows a scaling and non-negative shifting of the sum operation. Those additional nodes are considered in the template matching algorithm. Starting with the / node, the algorithm checks if there is a $*$ node. If true, it checks for a child $a$ which is greater or equal to 1 and a $+$ child. If there is no $*$ node, the algorithm checks if there is a $+$ node, if true, it checks for the sum and a non-negative child $b$ and continues as usual. Note that the scaling and shifting are optional. If there is neither a $*$ nor a $+$ node, the algorithm nevertheless finds a sum node if the template matches.

Using the generalized expression template, the next example of a convex function can be certified directly without further rearrangements. Again, this example is not feasible with CVX's implementation of DCP and no restriction of the domain is necessary. The expression reads

$$\sqrt{1 + \sum_{i=1}^{n} \exp(x_i)} \cdot \log\left(1 + \sum_{i=1}^{n} \exp(x_i)\right) = \text{sqrt}(1 + \text{sum}(\exp(x)))\log(1 + \text{sum}(\exp(x))),$$

with its Hessian

$$\frac{1}{\sqrt{1 + \text{sum}(\exp(x))}} \text{diag}(\exp(x))$$
$$+ \frac{\log\left(1 + \text{sum}(\exp(x))\right)}{2\sqrt{1 + \text{sum}(\exp(x))}} \left(\text{diag}(\exp(x)) - \exp(x)\exp(x)^{\top}/(2(1 + \text{sum}(\exp(x))))\right),$$

which matches the generalized expression template if $y$ is instantiated by vector(1), $z$ by $\exp(x)$, $a$ by 2 and $b$ by 1.