# OpenReview forum: "Convexity Certificates from Hessians"
_NeurIPS.cc/2022/Conference — NeurIPS 2022 Accept_

### Official Review · Reviewer_1deo · 2022-07-11

**Rating:** 8
**Confidence:** 5
**Soundness:** 3 good
**Presentation:** 3 good
**Contribution:** 3 good

**Summary:**

The authors present a framework to certify the convexity of some functions from their computational graphs. The usual tool to show the convexity of functions is to either show it by hand or use CVX which implements classical rules to know whether some compositions of functions are convex. As pointed out by the authors and as well known by the community, cvx has several limitations. In particular numerous functions need to be added manually as base "atoms" (i.e., basic convex functions) to demonstrate the convexity of some more complex functions. Even with these atoms, cvx may not be able to certify the convexity of some functions. The authors propose an alternative set of rules based on certifying the positive definiteness of Hessians through simple algebraic rules plus a few additional ones. The authors chose to present their ideas through several examples and a few propositions rather than a constructive framework. The authors present how their framework can encompass and surpass CVX to certify the convexity of new convex functions.

**Questions:**

- What is the definition of a normalized expression DAG? What is a normalized vectorized form?
- The authors mention a "language for representing multivariate function (see the supplementary material)" line 122 in the text. However, I do not see where this language is formally described. If the authors refer to Figure 1, this Figure is simply not understandable: what ['], [.] mean ? What is written in the factor line? The authors may come from a field where these expressions are obvious but if they wish their paper to be read they need to give proper definitions and gentler introductions to their formalism.
- The proof of Propositions 1 and 3 are rather trivial and could be put in the Appendix in order to give a formal definition of the framework or at least a pseudocode of the algorithm.
- An example motivated by real applications (such as the modeling of some data with a complex model) would greatly improve the relevance of the paper (though the current framework appears to be already a good tool for it).

**Limitations:**

- As the authors mention, the current formalism does not incorporate non-smooth functions which are often used as when using a ReLU or the l1 norm.
- The authors do not mention that convexity is not necessary to ensure convergence of e.g. first-order algorithms. Non-convex gradient dominating functions can also be optimized efficiently by a first-order method. This unfortunately limits the relevance of the approach for several problems that have been shown to satisfy a gradient dominating property while not convex.

**Strengths And Weaknesses:**

Strengths:
- The emergence of deep learning networks has diminished the appeal of convex modeling of problems. Yet, in numerous situations, practitioners may develop complex models (as in e.g. Bayesian statistics) where convexity could be a criterion to decide which model to prefer. So though the paper may seem not timely, I believe it can still be a very useful tool for the community.
- While the rules proposed by the authors appear extremely simple (Proposition 1) and not new, they are able to circumvent the need for all "patches" done in CVX (that is adding more and more basic functions). In other words, the framework proposed by the authors appears to be a better path to building a sound framework to prove the convexity of functions.
- The authors still require a few additional rules such as Proposition 2. But this might be where the paper appears the most interesting: finding as few additional and simple rules as possible to capture numerous cases beyond just adding log-sum-exp as an additional function.
- Overall, this is an original contribution to a topic that seems to have been often overlooked by either simply using CVX, or just checking the positive definiteness of the Hessian symbolically yet at low scale, or simply discarding the issue by saying that as long as the algorithm converges it could be enough.
- The available code is user-friendly.

Weaknesses:
While the idea is original and promising, there are several limitations, some can be seen as future directions, some are due in my opinion to the presentation.
- The main issue for me is the lack of formalism. While I appreciate the examples, there is no pseudo-code shared to understand the mechanisms of the approach and help peers to build upon it. The approach proposed by the authors may be a very interesting starting point to understanding the completeness/limitations of DAG representations of functions to certify convexity. Yet, without a few more formalism it cannot be built upon.
- There are also a few definitions that are simply not defined or understandable as detailed in the questions.

Update after having personally explored a bit more frameworks such as JAX from Google:
- Though again, I would have liked to understand some pseudocode better to help other people build on this idea, I also realized that this could probably easily be implemented as part of homoiconic programming languages such as Julia or JAX. Namely, these languages parse functions in form of expressions which I think are quite similar to what the authors are doing. So I updated my score since I believe this is an original yet simple and really useful tool that the authors developed and that could readily be integrated into some machine learning packages. I would sincerely appreciate it if the authors could add more background (in the appendix with some pseudocode or some generic framework *if possible*), this could have the potential for further research maybe as long as the authors make the effort to help other researchers build on it.

---

> ### Author Response · Authors · 2022-07-28
> **Answers to questions and comments**
>
> Thank you for your helpful comments and questions that we address in the following.
>
> - *The main issue for me is the lack of formalism. While I appreciate the examples, there is no pseudocode shared to understand the mechanisms of the approach and help peers to build upon it.*
>
> Now, we have also added pseudocode to the supplemental material. As you will see, the algorithm is not very difficult. The difficult part was to come up with the template and to realize how powerful it is. It is still surprising to us that the basic algorithm plus a single template is sufficient to cover all differentiable atoms of CVX. We plan to make the Web interface and the code of our implementation publicly available.
>
> - *What is the definition of a normalized expression DAG? What is a normalized vectorized form?*
>
> Vectorized simply means that the expressions are expressed without indices. Instead of indices, type information is provided. For instance, for the expression $x^\top Ax$ the information that $x$ is a vector and $A$ is a matrix must be provided.
>
> In a normalized expression, all common sub-expressions have been determined and now appear exactly once in the computational graph (DAG). For example, the DAG for the expression $x^\top Ax + \exp(y^\top Ax)$ has only one node for $x$ and only one node for $Ax$. In the supplemental material, we now explain how to get normalized expression DAGs.
>
> - *The authors mention a "language for representing multivariate function (see the supplementary material)" line 122 in the text. However, I do not see where this language is formally described. If the authors refer to Figure 1, this figure is simply not understandable: what ['], [.] mean? What is written in the factor line? The authors may come from a field where these expressions are obvious, but if they wish their paper to be read, they need to give proper definitions and gentler introductions to their formalism.*
>
> Indeed, Figure 1 provides the grammar for our language in standard EBNF notation. We now clarify in the supplemental material that our formal language builds on standard semantics, where ['] means transpose and a preceding [.] means elementwise operation. We also provide some examples to illustrate the language.
>
> - *The proof of Propositions 1 and 3 are rather trivial and could be put in the Appendix in order to give a formal definition of the framework or at least a pseudocode of the algorithm.*
>
> Indeed, the non-trivial challenge was finding the template in Proposition 2. However, we would prefer to keep both propositions and their proofs in the main text, because Proposition 1 completely describes how the Hessian approach relates to the DCP approach in the differentiable case, and Proposition 2 (and its short proof) introduces the psd template, which is really important for the Hessian approach.

---

> > ### Author Response · Authors · 2022-08-03
> > **Small Update on Integration with Julia and JAX**
> >
> > As the reviewer correctly points out, our approach can readily be integrated into frameworks/languages like Julia and JAX, but also PyTorch and TensorFlow. All frameworks support computing derivatives and store an internal representation in the form of an expression/computation tree. The proposed algorithm (now with pseudo-code in the appendix, as suggested by the reviewer) can then be used here as well. We did incorporate the algorithm into one similar framework (details are not revealed for anonymity reasons.)

---

> > ### Comment · Reviewer_1deo · 2022-08-04
> > **Answer to authors**
> >
> > I thank the authors for their detailed answers to my comments that they addressed very well in the updated version of the paper.
> >
> > I have a few additional comments for making the better a bit better:
> > - The pseudocode is great! However, I'm sincerely sorry, but there are still some unclear zones: what is the procedure combine intervals? How do the authors incorporate rule 4 of Proposition 1? A bit more explanation would be welcome.
> > - Thank you for explaining the notion of normalized graphs, I did not know about it and it is very interesting to me. Do the authors have a reference for it? Moreover, I would assume that going from Fig. 2 to Fig. 3 in the supplementary material is done by some well-known algorithm, would it be possible to mention it, or is it just trivial?
> >
> > Overall I sincerely believe that this is a truly original yet simple paper that provides a very useful and important tool for modeling purposes. With the additional pseudocode, it can be a basis for future work that has not been explored yet and may let people rethink such problems. Finally, as I mentioned, this framework should easily be incorporated into modern machine learning frameworks that parse expressions in a way that the proposed algorithm could use.
> >
> > The comments of Reviewer yPQf have been fully addressed. About the comment "CVX can also solve problems": the current paper is more flexible in my opinion and, again, it could easily be incorporated into numerous languages while offering the possibility for the user to use their favorite on-the-shelf solver rather than predefined ones. The Hessian approach is clearly more powerful than DCP and the computational complexity is clearly extremely small a priori for modeling purposes since it works on normalized DAGs.
> >
> > I have updated my score to defend the paper to foster discussions on this paper.

---

> > > ### Author Response · Authors · 2022-08-04
> > > **Answer to Reviewer 1deo**
> > >
> > > Thank you for the encouraging feedback and your questions that we answer below.
> > >
> > > - *The pseudocode is great! However, I'm sincerely sorry, but there are still some unclear zones: what is the procedure combine intervals? How do the authors incorporate rule 4 of Proposition 1? A bit more explanation would be welcome.*
> > >
> > >  In order to highlight the key algorithmic ideas, we kept the pseudocode somewhat simpler than the actual implementation. However, we should have mentioned that for matching psd Rule 4, we also use the tree matching algorithm at multiplication nodes for the simple template given by Rule 4 (analogously to matching the complex psd expression template). We use the tree matching algorithm only if one of the arguments of the multiplication node is not of type scalar. Otherwise, if one of the arguments is of type scalar, then we use the interval arithmetic. Actually, the high-level description in lines 257-269 in the main paper is more precise about this and states that Rule 4 and the psd expression template both need special treatment (tree matching).
> > >
> > > - *Do the authors have a reference for it? Moreover, I would assume that going from Fig. 2 to Fig. 3 in the supplementary material is done by some well-known algorithm, would it be possible to mention it, or is it just trivial?*
> > >
> > > This is indeed part of the folklore in compiler construction. Going from Figure 2 to Figure 3 can be done similarly to our certification algorithm by traversing the original expression tree (Figure 2) and replacing subexpressions that have been encountered before by links to their expression DAGs. See for example the book by Steven Muchnick on Advanced Compiler Design and Implementation, or the classical text by Alfred Aho, Monica Lam, Ravi Sethi, and Jeffrey Ullman on Compilers: Principles, Techniques, and Tools.

---

### Official Review · Reviewer_WYiR · 2022-07-12

**Rating:** 5
**Confidence:** 1
**Soundness:** 3 good
**Presentation:** 3 good
**Contribution:** 3 good

**Summary:**

The paper implements the Hessian approach for certifying the convexity of functions, which complements the well-established disciplined convex programming approach and works for non-differentiable functions.

**Questions:**

See the comments above.

**Limitations:**

Yes.

**Strengths And Weaknesses:**

Strengths:
- The paper writing is very good and the presentation is clear.
- The implementation is based on a formal language that is sufficiently rich to support a large class of problems. The anonymized implementation demo is promising.
- The fundamentals of the implementation seem to be solid by using computational graphs for Hessians, the rule set and atoms.

I would say I am not an expert on this field, i.e., computational frameworks for convex optimization. My best educated guess is that the paper has a solid background as well as an attractive web demo. I am willing to learn from other reviewers' comments and the rebuttal.

---

> ### Author Response · Authors · 2022-07-28
> **Thank you**
>
> Thank you for the positive feedback. We hope that our rebuttal helps you with your decision.

---

### Official Review · Reviewer_yPQf · 2022-07-12

**Rating:** 5
**Confidence:** 4
**Soundness:** 3 good
**Presentation:** 3 good
**Contribution:** 2 fair

**Summary:**

This work presents an implementation of the Hessian approach for certifying the convexity of differential functions. It is argued that it complements the well-established disciplined convex programming (DCP). Some examples are presented to show that the proposed approach can be even more powerful than the DCP approach (e.g., in Section 7).

**Questions:**

It would be better to clarify the computational properties of the proposal Hessian approach.

**Limitations:**

The evaluation of the proposed Hessian approach is relatively limited and not solid. More practical examples will help convince the effectiveness.

**Strengths And Weaknesses:**

Strengths:
+ The paper is in general well-written, and the main message is easy to follow.
+ The analysis is solid and correct, and the unique contributions are highlighted in a good way (unlike a few other Neurips submission I reviewed, which has obvious technical flaws or unclear contributions).
+ An implementation of the proposed approach is provided on an anonymized website.

Weakness
- Although well-motivated, it is unclear about the efficiency in terms of computation and storage to use the Hessian approach. Normally, people tend to avoid computing the Hessian explicitly, let alone verifying the positive semidefinitness of the Hessian over the entire domain. It seems like the proposed implementation can only deal with lower-dimensional problems or problems with very simple structures. What is the computational scalability in terms of time and storage?
- It is not very clear why the Hessian approach is better than the well-established DCP. For one thing, the DCP is more flexible (by simply adding more atoms) and requires less computation (without toughing the complicated Hessian computation). For another thing, the DCP is not just testing convexity, and it can further return a standard convex optimization (in a conic form) for practical computation. It seems that the Hessian approach can only test convexity in a large sense.
- From a technical-wise perspective, Proposition 1 is just another way to express the DCP rules for differential functions. It is unclear about differences in terms of the computational graphs as DAGs, compared to the DCP way. How do the DAGs work for a standard log det function, i.e. $f(X) = -\log \det(X)$?

---

> ### Author Response · Authors · 2022-07-28
> **Answers to questions and comments**
>
> Thank you for your helpful comments and questions that we address in the following.
>
> - *Although well-motivated, it is unclear about the efficiency in terms of computation and storage to use the Hessian approach. Normally, people tend to avoid computing the Hessian explicitly, let alone verifying the positive semidefinitness of the Hessian over the entire domain. It seems like the proposed implementation can only deal with lower-dimensional problems or problems with very simple structures. What is the computational scalability in terms of time and storage?*
>
> It is important to note that we compute *symbolic* Hessians. The size of a symbolic Hessian is a small constant times the size of the input expression, that is, typically only a few hundred bytes. Notably, the size of the expression does not grow with the problem size. For example, the Hessian of the expression $x^\top Ax$,  for a symmetric matrix $A$, is $2A$, which can be encoded by two ASCII characters (two Bytes) and is independent of any problem-specific instantiation of $A$.
>
> The symbolic Hessian can be computed in time linear in the size of the input expression by reverse mode autodiff, and our algorithm for certifying that a Hessian is psd is linear in the size of the *symbolic* expression for the Hessian. For all examples in the paper, the running time of the algorithm is in the range of milliseconds, *independent* of the dimension of the variables. We now add running time experiments to the supplemental material.
>
> - *It is not very clear why the Hessian approach is better than the well-established DCP. For one thing, the DCP is more flexible (by simply adding more atoms) and requires less computation (without touching the complicated Hessian computation).*
>
> Actually, the DCP approach is less flexible. Note that it needs a new symbol for every new atom, while the Hessian approach can certify them on the natural input language. Furthermore, new DCP atoms also need to be certified at some point. Neither approach is better than the other. Both have complementary strengths and weaknesses. One strength of the Hessian is its flexibility, in the sense that it works on a natural input language. Therefore, it can be used to certify new DCP atoms.
>
> - *For another thing, the DCP is not just testing convexity, and it can further return a standard convex optimization (in a conic form) for practical computation. It seems that the Hessian approach can only test convexity in a large sense.*
>
> DCP also only certifies convexity. CVX, which DCP is a part of,  can transform a convex input problem into a problem in standard form, which then can be solved by a standard solver. The Hessian approach can also be used with standard solvers like IPOPT, Knitro, or GENO that avoids CVX's often inefficient problem transformation into standard form. Note that the latter solvers do not need the Hessian to solve the problem. The symbolic Hessian is only needed to certify convexity.
>
> - *From a technical-wise perspective, Proposition 1 is just another way to express the DCP rules for differential functions."*
>
> Actually, Proposition 1 proves that in the differentiable case, the Hessian approach is always at least as powerful as the DCP approach. The examples in Section 7 and in the supplemental material show that it is actually more powerful than CVX's implementation of the DCP approach.
>
> - *It is unclear about differences in terms of the computational graphs as DAGs, compared to the DCP way. How do the DAGs work for a standard log det function, i.e. log(det(X)?*
>
> We are not sure what is meant by this question. The computational DAG of $\log(\det(X))$ is just $\log \rightarrow \det \rightarrow X$. Its Hessian is $-{X^{-1}}^\top \otimes X^{-1}$. Note that the DCP approach, and thus the expression graphs in DCP, need a new symbol for logdet expressions and cannot just use the symbols for $\log$ and $\det$. In CVX this atom is called log_det.
>
> - *It would be better to clarify the computational properties of the proposal Hessian approach.*
>
> We have added pseudocode to the supplemental material.

---

> ### Comment · Reviewer_yPQf · 2022-08-07
> **Computational efficiency of the proposed Hessain approach**
>
> I want to thank the authors for the response. However, the response does not address my concerns about the Hessian approach's computational efficiency and storage requirement. I don't see why reviewer 1deo claims that ``The comments of Reviewer yPQf have been fully addressed.'' This arbitrary claim makes me suspicious about the other judgments.
>
> Again, I maintain my original review. The paper is generally well-written and well-motivated, but
> - "it is unclear about the efficiency in terms of computation and storage to use the Hessian approach. Normally, people tend to avoid computing the Hessian explicitly, let alone verifying the positive semidefinitness of the Hessian over the entire domain. It seems like the proposed implementation can only deal with lower-dimensional problems or problems with very simple structures."
>
> The authors argue this question from the perspective of symbolic Hessians. Don't the symbolic computation make efficiency worse? For very simple and small-scale problems, symbolic computations can be very easy and efficient, such as MATLAB and MATHEMATICA (actually, this makes me wonder how the proposed approach compares with the symbolic computation in MATHEMATICA). However, it is a better choice to avoid symbolic computation as much as possible in practice.  Table 1 in the supplementary materials only has very simple examples, which do not justify or support the computational futures.

---

> > ### Author Response · Authors · 2022-08-08
> > **Computational Efficiency of Computing Hessians**
> >
> > Thank you for bringing up your concerns about the efficiency of the Hessian computation that we address below. We hope that our answers remove any concerns about the efficiency of our approach
> >
> > - *Don't the symbolic computation make efficiency worse?*
> >
> > No, it does not. The size of the symbolic Hessian only grows linearly in the size of the input expression, while the Hessian instantiated on problem data can grow quadratically in the size of the input data. In machine learning the input expression typically is small, at most a few hundred bytes, while the input data can be very large (gigabytes or more). Actually, it has been *proven*  mathematically that the size of the Hessian is *always* at most a *constant* times the size of the input expression, see for example [1].
> >
> > Here are the details:  Our approach, as done similarly by TensorFlow, PyTorch, or JAX, considers input expressions as DAGs. When computing the derivative, we go through the computation graph, and for each operator node, we compute the partial derivative of this node and add it to the computation graph (DAG). (For instance, for 'sin' we add 'cos' and a 'times' operation.) Obviously, this only adds a *small constant* for each operator node and hence, a small constant for the whole DAG. This small constant is usually between 2 and 3 in practice and it has been *proven* in [1, Chapter 4, Memory Issues and Complexity Bounds] that it is always at most 5. Now computing the second derivative, one has to perform this operation twice in a row; once on the original input expression and then on the DAG of the first derivative. Hence, the size of the computation DAG of the Hessian ( = the size of the symbolic expression) is proven to be at most 25 times the input expression but usually between 4 and 9 times the input expression. Hence, it will always only be only a *constant* times the size of the input DAG, that is, usually a few hundred bytes. All our approach needs is this symbolic computation DAG.
> >
> > Maybe the confusion comes from the fact that we do not need to *evaluate* the computation DAG. Evaluating the DAG, that is, performing the individual computations that are stored in the DAG can be time-consuming and of course, depends on the dimension of the problem. But our approach does not need to do this.
> >
> > - *For very simple and small-scale problems, symbolic computations can be very easy and efficient, such as MATLAB and MATHEMATICA (actually, this makes me wonder how the proposed approach compares with the symbolic computation in MATHEMATICA).*
> >
> > Note that we build on a recent result in computing *vectorized* symbolic derivatives [2], which is based on reverse mode autodiif and comes with the efficiency guarantees that we have described above. MATLAB and MATHEMATICA, so far, do not compute (symbolic) derivatives in vectorized form, that is, they need indices to write down the derivatives. They do not operate on the compact representation that our approach uses. (For example, our representation/approach is *independent* of the problem dimension, which  MATLAB or MATHEMATICA is not). While derivatives with indices can still be written down succinctly, they are much harder to analyze. We do not have this issue in our approach. Hence, our representation and approach are much more efficient than the output that comes from MATLAB or MATHEMATICA.
> >
> > - *However, it is a better choice to avoid symbolic computation as much as possible in practice.*
> >
> > This is a myth. This claim was made when comparing symbolic with automatic differentiation. However, there is no difference here, because we use, as discussed above, reverse mode autodiff to compute the symbolic derivatives.
> >
> >
> > [1] Andreas Griewank and Andrea Walther. Evaluating Derivatives: Principles and Techniques of Algorithmic Differentiation, SIAM 2008.
> >
> > [2] Sören Laue, Matthias Mitterreiter, Joachim Giesen. Computing Higher Order Derivatives of Matrix and Tensor Expressions. NeurIPS 2018.

---

> > > ### Author Response · Authors · 2022-08-09
> > > **Proof for efficiency**
> > >
> > > Dear reviewer yPQf,
> > >
> > > did the mathematical proof from the book [1, chapter 4] convince you that computing the Hessian DAG is very efficient and can be done in linear time and space? If not, please let us know. Thank you!

---

### Meta-Review · Area_Chair_85KC · 2022-08-25

**Recommendation:** Accept
**Confidence:** Less certain

**Metareview:**

This was a though call. Two reviewers were only slightly positive about the paper and one of them did not seem to understand it very much, while a third reviewer who is an expert, really like the paper very much and so I tend to accept.

**Award:**

No

---

### Decision · Program_Chairs · 2022-09-14

Accept